# Healthy Diet-Related Knowledge, Attitude, and Practice (KAP) and Related Socio-Demographic Characteristics among Middle-Aged and Older Adults: A Cross-Sectional Survey in Southwest China

**DOI:** 10.3390/nu16060869

**Published:** 2024-03-17

**Authors:** Lin Fu, Ya Shi, Shengping Li, Ke Jiang, Laixi Zhang, Yaqi Wen, Zumin Shi, Yong Zhao

**Affiliations:** 1School of Public Health, Chongqing Medical University, Chongqing 400016, China; 2023121648@stu.cqmu.edu.cn (L.F.); 2023121694@stu.cqmu.edu.cn (Y.S.); 2021110631@stu.cqmu.edu.cn (K.J.); 2021120819@stu.cqmu.edu.cn (L.Z.); wenyaqi@stu.cqmu.edu.cn (Y.W.); 2Research Center for Medicine and Social Development, Chongqing Medical University, Chongqing 400016, China; 3Research Center for Public Health Security, Chongqing Medical University, Chongqing 400016, China; 4Nutrition Innovation Platform-Sichuan and Chongqing, School of Public Health, Chongqing Medical University, Chongqing 400016, China; 5Chongqing Health Center for Women and Children/Women and Children’s Hospital of Chongqing Medical University, Chongqing 400012, China; 2019110981@stu.cqmu.edu.cn; 6Human Nutrition Department, College of Health Sciences, QU Health, Qatar University, Doha 2713, Qatar; zumin@qu.edu.qa; 7Chongqing Key Laboratory of Child Nutrition and Health, Children’s Hospital of Chongqing Medical University, Chongqing 400014, China

**Keywords:** middle-aged and older adults, healthy diet, knowledge, attitude, practice

## Abstract

Objective This study aimed to investigate the current status and influencing factors of healthy diet knowledge, attitude, and practice (KAP) among middle-aged and older adults aged 45–75 in Southwest China. Methods A questionnaire survey was conducted among 1822 middle-aged and older adults in Southwest China (including Guizhou, Sichuan, Yunnan, and Chongqing) from February to May 2021. Results The average score of healthy diet knowledge was (4.82 ± 2.98), with a passing rate of 7.6%. The mean score of healthy diet attitude was (21.26 ± 4.18), with a passing rate of 69.5%. The average score of healthy diet practice was (13.76 ± 2.84), with a passing rate of 55.5%. The score for healthy diet KAP was (39.85 ± 7.21), with a passing rate of 41.2%. Univariate analysis showed that the scores of healthy diet KAP were significantly different among participants of different ages, genders, ethnicities, residences, education levels, monthly household incomes, and regions, as well as varying according to whether several generations have lived in the same area (*p* < 0.05). The results of multiple linear regression showed that the healthy diet KAP of participants was influenced by age, gender, residence, education level, monthly household income, and region (*p* < 0.05). Conclusion The healthy diet KAP of middle-aged and older adults aged 45–75 in Southwest China shows room for improvement. The knowledge regarding healthy diet was relatively low, and certain specific healthy diet practices were not up to the standard. However, there was a positive trend in the attitude towards a healthy diet. Healthy diet education should be promoted for middle-aged and older adults.

## 1. Introduction

Global population aging and the extension of life expectancy have placed a considerable burden on healthcare systems [1]. People older than 60 years could account for half of the disease burden in high-income countries [2]. The rapid aging of the population poses a major challenge to public health in China. According to the 2020 population census, there are currently 264 million people (accounting for 18.7% of the total population) aged 60 and above. It is projected that by 2040, 28% of the population will be aged 60 and above [3]. Middle-aged and older adults play a vital role in society, and their health directly affects the overall well-being of the community [4,5,6]. With increasing life expectancy, chronic noncommunicable diseases remain the primary global contributors to the disease burden and to disability among middle-aged and older adults [6,7,8]. According to data from the National Health Commission of China, as of the end of 2018, 150 million people (nearly 90% of older adults) were affected by chronic diseases [9]. Diet behavior is a factor that profoundly influences the health outcomes of this population group [10,11]. Unhealthy eating behaviors, poor dietary status, and imbalanced food choices are modifiable risk factors for various chronic diseases, including cardiovascular diseases (CVDs) and Type 2 Diabetes Mellitus (T2DM) [12,13]. Conversely, maintaining healthy diet habits can prevent age-related chronic diseases and promote healthy aging [14,15]. During the era of personalized nutrition, the global trend towards adopting westernized dietary patterns reflects a deficiency in nutritional health knowledge, leading to a rise in major chronic diseases within certain populations and hindering the promotion of healthy aging [16].

In southwestern China, factors such as geographical environment contribute to a preference for high-fat, high-salt, and spicy foods, which may increase the risk of chronic diseases [17,18]. Furthermore, the prevalence of chronic non-communicable diseases, such as diabetes, cardiovascular diseases, and obesity, is relatively high in southwestern China [19,20,21]. Previous research has shown that individuals’ diet knowledge influences their food choices, which in turn are influenced by their attitudes toward health [22]. Therefore, to understand the healthy diet status of middle-aged and older adults, it is important to pay attention to their healthy diet-related knowledge, attitudes, and practices.

The utilization of the KAP theoretical model is prevalent in research and studies examining diet behavior changes and public health issues. Based on the Knowledge–Attitude–Practice (KAP) theoretical model [23], the healthy diet-related Knowledge–Attitude–Practice (KAP) model among middle-aged and older adults in southwestern China explains how individual healthy diet behaviors follow a process comprising three stages: acquiring knowledge about healthy diets, developing attitudes towards healthy diets, and engaging in healthy diet practices. Owing to the lack of healthy diet knowledge and the prevalence of attitude deviations and inadequate practices, there is much room for improvement in the healthy diet behaviors of middle-aged and older adults [24]. A previous study by Yang et al. [25] reported that Chinese adults exhibit poor performance in terms of diet knowledge, attitudes, and behaviors. A related study also showed that older adults with lower diet-related knowledge tend to have poorer practices related to diet [26]. Overall, older adults often have insufficient knowledge regarding healthy diets, especially when compared with younger individuals [27,28]. Having higher levels of knowledge and more positive attitudes toward a healthy diet are associated with positive diet behaviors, including increased consumption of fruits and vegetables and actively seeking out nutritional information [25]. While knowledge alone is insufficient to generate remarkable changes in behavior, improving knowledge is crucial for attitude transformation and establishing the capability for change [29]. Investigating and understanding the healthy diet-related knowledge, attitudes, and practices of middle-aged and older adults is an important step in providing information for effective behavior change interventions and evaluating healthy diet-related education programs.

Therefore, this study aims to investigate the knowledge, attitudes, and practices (KAP) related to a healthy diet among middle-aged and older adults in southwestern China. Specifically, this study will explore the factors influencing knowledge about healthy diet, attitude towards diet choices, and actual diet practices. The intended outcomes of this research are to identify potential gaps in knowledge, understand the factors influencing diet behaviors, and provide insight for developing targeted interventions to promote a healthier diet among this population.

## 2. Materials and Methods

### 2.1. Schematic Representation of the Cross-Sectional Studies

The survey on healthy diet-related knowledge, attitudes, and practices (KAP) was initiated in 2021, and data were collected from middle-aged and older adults aged 45–75 residing in Southwest China. The aim of this study was to investigate the current status of healthy diet-related KAP and identify the factors influencing it. To ensure the study’s objectives were met, specific inclusion and exclusion criteria were established for the study population. The minimum sample size was determined based on data from the Chinese Adult Health KAP survey. The study utilized a questionnaire developed by experts in science communication and popularization from the Chinese Society of Nutrition. Data collection was carried out according to a planned schedule of face-to-face on-site questionnaires, with subsequent data processing and analysis (See Figure 1).

### 2.2. Study Design and Sample

A cross-sectional survey was conducted between February and May 2021 in Southwest China (Sichuan, Chongqing, Yunnan, and Guizhou). The sample size required for the study was estimated by the sample size calculation formula of the cross-sectional study, as follows: $n=\frac{{Z_{\alpha }}^{2}\;\times \;pq}{d^{2}}$. A previous study showed that the diet-related knowledge rate among Chinese adults was 14.7% [30]. Therefore, we set *p* = 0.147, *d* was 0.15*p*, and the confidence level was 0.95; a sample size of 1035 could be calculated by PASS 15.0 (NCSS, LLC, Kaysville, UT, USA). Considering the 20% non-response rate, the sample size was at least 1294 participants.

### 2.3. Inclusion and Exclusion Criteria

The following inclusion criteria were used: (1) aged 45–75 years, (2) ability to understand and complete questionnaires independently, and (3) informed consent and voluntary participation in the survey. The following exclusion criteria were used: (1) unwillingness to participate in this survey and (2) inability to complete the survey due to illness or other reasons.

A total of 1843 middle-aged and older adults participated in our study. After excluding outliers and missing values, 1822 participants were included in the analysis. The study was reviewed and approved by the Ethics Committee of Chongqing Medical University (approval number: 2021041). The participants provided their written informed consent to participate in this study, and all of them signed informed consent before the investigation.

### 2.4. Data Collection

The questionnaire was designed by the Chinese Nutrition Society Science Communication and Popularization Experts. In addition, the survey was a national food culture survey conducted by the Chinese Nutrition Society. The questionnaire consisted of four parts, including basic demographic data, the healthy diet-related knowledge questionnaire, the healthy diet-related attitude questionnaire, and the healthy diet-related practice questionnaire.

The first part covered basic demographic data, including gender, age, ethnicity, residence, monthly household income, whether generations have lived in the area, education level, and body mass index (BMI). Monthly household income was divided into low (<5000 RMB), medium (5000–9999 RMB), and high (≥10,000 RMB). Education level was classified into basic level (junior high school and below), secondary level (high school/secondary school/technical school/junior college), and higher level (bachelor’s degree and above). Body mass index (BMI) was calculated by self-reported height and weight (weight/height^2^) and divided into underweight (<18.5 kg/m^2^), normal (18.5 kg/m^2^ ≤ BMI < 24 kg/m^2^), overweight (24 kg/m^2^ ≤ BMI < 28 kg/m^2^), and obese (BMI ≥ 28 kg/m^2^) [31]. Please refer to Appendix A for detailed questions regarding the KAP of healthy diet. The second part comprised the healthy diet-related knowledge questionnaire. This section consisted of 16 questions aimed at assessing participants’ knowledge of healthy diet habits, dietary guidelines, and the benefits of a balanced diet. These questions covered food types, portions, nutritional requirements, and the impact of diet choices on health, and each question was scored as 1 point for a correct answer and 0 points for an incorrect answer. The total score of this section ranged from 0 to 16, and a high score indicated a high knowledge level on the topic. The third part comprised the healthy diet-related attitude questionnaire. In this section, participants were asked to express their attitudes and beliefs regarding different aspects of a healthy diet. These questions explored their views on the importance of nutrition, motivations for adopting healthy diet habits, and so on. This section consisted of 16 questions. “Agree” was scored as 2 points, “Neutral” scored as 1 point, and “Disagree” scored as 0 point. Moreover, reverse questions were combined to assign reverse scores. The total score for the attitude section ranged from 0 to 32, with higher scores meaning more positive attitudes. The fourth part comprised the healthy diet-related practice questionnaire. This section gathered information about participants’ actual diet practices. The survey content included daily food choices, cooking methods, frequency of dining out, and more. This section consisted of 11 questions, which were scored positively and negatively, similar to the third section. The total score for this section ranged from 0 to 22, and higher scores indicated good healthy diet practices. The total score of the KAP questionnaire was the sum of the scores from the knowledge, attitude, and practice sections, with a maximum score of 70. It was hypothesized that a KAP score greater than or equal to 60% of the total score (42 points) indicated a passing score, indicating that the individual’s KAP score related to healthy diet was at or above a satisfactory level.

Additionally, the pre-survey was conducted in February 2021 using the “Questionnaire-Star” platform, and a total of 1059 questionnaires were collected, the Cronbach’s alpha coefficient was 0.825, indicating that the questionnaire had acceptable reliability. The Kaiser–Meyer–Olkin (KMO) measure was 0.859, indicating that the questionnaire had acceptable internal consistency and structural validity. Then, the questionnaire was revised and discussed over several rounds by experts in the fields of nutrition and sociology; finally, the content validity of the questionnaire was considered acceptable.

### 2.5. Statistical Analysis

SPSS 25.0 (IBM Corp., Armonk, NY, USA) was used for statistical analyses. Normal distribution measurement data were described by mean and standard deviation ($\bar{x}\pm s$). Differences between groups were tested by the two independent-sample *t*-test and one-way ANOVA. The Bonferroni test was used for multiple comparisons in the ANOVA. Multivariable linear regression was used to assess the determinants of KAP. The data utilized in the multiple linear regression analysis was assessed and met the assumptions required for conducting the analysis. A *p* value < 0.05 was considered statistically significant.

## 3. Results

### 3.1. Participants’ Characteristics

Among the 1822 participants in this study, the majority of participants were in the age range of 45–59 (81.2%), and more than half were male (50.2%). The Hans (91.7%) comprised the highest number of participants. Over half of the participants’ households were in urban areas (58.8%), with a high percentage of participants whose families have been living in the area for generations (81.8%). The highest number of participants had a lower monthly family income (52.2%), followed by medium income (24.8%), and then a higher income (22.9%). Among the 1822 participants, 33.2% were overweight and 6.8% were obese (see Table 1).

### 3.2. Status of Healthy Diet-Related Knowledge, Attitudes, and Practices (KAP)

The healthy diet KAP score of the 1822 participants was (39.85 ± 7.21), with a passing rate of 41.2% (751/1822). Among them, the healthy diet knowledge score was (4.82 ± 2.98), with a passing rate of 7.6% (138/1822). Significant differences existed in the healthy diet knowledge score among different age groups, with the 45–59 age group scoring higher than the ≥60 age group (*p* < 0.05). The scores of the 45–59 age group and ≥60 age group were (4.94 ± 3.01) and (4.30 ± 2.82), respectively. Differences existed in the healthy diet knowledge score among different genders (*p* < 0.05). The scores of males and females were, respectively, (4.63 ± 2.89) and (5.01 ± 3.06). The healthy diet knowledge score of the participants living in urban areas was (5.42 ± 3.08), higher than that of the participants living in rural areas (3.96 ± 2.60); notably, the difference was statistically significant (*p* < 0.05).

The healthy diet attitude score was (21.26 ± 4.18), with a passing rate of 69.5% (1266/1822). Significant differences existed in healthy diet attitude scores among different age groups, with the 45–59 age group scoring higher than the ≥60 age group (*p* < 0.05). The score of the participants living in urban areas was (21.92 ± 3.89), which was higher than that of the participants living in rural areas (20.32 ± 4.38); notably, the difference was statistically significant (*p* < 0.05).

The score of healthy diet practices was 13.76 ± 2.84, and the passing rate was 55.5% (1011/1822). The score of females was (14.19 ± 2.69), which was higher than that of males (13.33 ± 2.92); notably, the difference was statistically significant (*p* < 0.05). The healthy diet practice score of the participants living in urban areas was (14.18 ± 2.84), which was higher than that of the participants living in rural areas (13.17 ± 2.74); the difference was statistically significant (*p* < 0.05) (see Table 2).

### 3.3. Univariate Analysis of Healthy Diet-Related Knowledge–Attitude–Practice (KAP)

The results of the univariate analysis showed that the score of healthy diet KAP was significantly different among participants of different age groups, genders, ethnicities, residences, education levels, monthly household incomes, and regions, as well as according to whether generations have lived in the same area (*p* < 0.05) (see Figure 2).

### 3.4. Multiple Linear Regressions to Identify Factors Affecting the Healthy Diet KAP

To further explore the influencing factors on the score of healthy diet KAP, factors with statistically significant differences in univariate analysis were taken as independent variables, and the score of healthy diet KAP as the dependent variable. The results of multiple linear regression showed that the healthy diet KAP score of middle-aged and older adults was influenced by age, gender, residence, education level, monthly household income, and region (*p* < 0.05). The KAP score in the ≥60 age group was lower than in the 45–59 age group (*p* < 0.05). Compared with males, females had higher scores for healthy diet KAP (*p* < 0.001). Those living in urban areas had higher scores for healthy diet KAP than those living in rural areas (*p* < 0.001). Compared with those with basic education, those who had higher education had higher scores for healthy diet KAP (*p* < 0.001). Compared with middle-aged and older adults who lived in Guizhou, those who lived in Chongqing, Sichuan, and Yunnan had higher scores for healthy diet KAP (see Table 3).

## 4. Discussion

The KAP theoretical model is frequently utilized in investigations and studies concerning diet behavior changes and public health issues [32]. Based on the fundamental principles of KAP theory, enhancing knowledge can bring about changes in attitudes and behaviors, consequently alleviating the burden of diseases [33,34]. Previous research has extensively examined the correlation between diet and health status among older adults [35]. Studies have shown that healthy diet plays a crucial role in promoting health and preventing diseases in this population [36]. However, in southwestern China, the diet behavior of middle-aged and older adults has not been adequately identified. Hence, we employed the KAP theoretical model to evaluate the healthy diet situation of the middle-aged and older adults in southwestern China. This study represents an important step toward gaining a better understanding of the healthy diet patterns and behaviors of this population and will contribute to a more in-depth assessment of the impact of diet on disease development in the future.

In this study, the overall passing rate of healthy diet KAP among middle-aged and older adults in southwestern China was only 41.2%. Specifically, the passing rate for healthy diet knowledge was as low as 7.6%. Particularly, the accuracy of specific healthy diet knowledge, such as the recommended intake of cooking oil and salt per day for humans, was relatively low, which might be related to the preference for high-salt and high-oil foods in southwestern China. Regarding healthy diet attitudes, the passing rate was 69.5%, indicating positive attitudes toward healthy nutrition among middle-aged and older adults in southwestern China. However, the passing rate for healthy diet practice was only 55.5%. Regarding specific healthy diet behaviors, 21.1% of the respondents never separate raw and cooked ingredients, and only 32.5% pay attention to nutritional labels when purchasing pre-packaged foods. This finding indicates that the situation of healthy diet practices among middle-aged and older adults in southwestern China remains pessimistic. A cross-sectional study showed that although most residents had a positive attitude toward nutrition labeling, the awareness rate of nutrition labeling was only 32.7%, and the utilization rate was 38.5% [37]. Therefore, further strengthening healthy diet education for middle-aged and older adults, improving their awareness of and attention to healthy diets, and promoting the development of healthy diet habits are necessary.

Different genders of the study participants showed differences in scores for healthy diet knowledge and healthy diet practice, with females scoring higher than males in both aspects (refer to Table 2); this finding was in line with the results from other research [38]. This finding may have resulted from women being assigned additional responsibilities for cooking and diet decisions in society. In fact, in most Chinese households, women take on the role of cooking, and they have a greater focus on healthy diet knowledge, such as salt intake levels [39]. Accordingly, they may pay extra attention to the healthy diets of family members and possess further diet knowledge and skills. Furthermore, relevant studies [40,41] have indicated that differences might exist between males and females in diet needs, preferences, and health awareness, with females placing high emphasis on the nutrition and health aspects of diet and paying extra attention to balance and diversity. Thus, it is important to develop gender-specific educational programs that address the unique dietary needs and health awareness of different genders. Meanwhile, it is important to raise awareness of and advocate for the importance of shared responsibilities for healthy diets within households through promotional activities and campaigns, emphasizing the involvement of males in cooking and decision-making processes with regard to diets. Previous research has shown a positive correlation between socioeconomic status and dietary quality [42]. In this study, the participants residing in urban areas scored higher in healthy diet knowledge, attitudes, and practices than those residing in rural areas (Refer to Table 2), which was in line with the findings of other studies conducted in China [25,43]. This finding may be related to the imbalance in urban and rural development in China, where urban areas typically have high socioeconomic levels, enabling residents to easily purchase high-quality and healthy ingredients, and offer other opportunities to engage in healthy diet behaviors. Therefore, it is recommended to prioritize the implementation of intervention measures and policies to promote access to high-quality and healthy food ingredients in rural areas, as well as provide opportunities for rural residents to engage in healthy dietary behaviors.

In this study, significant differences were found in healthy diet KAP scores among different ages, genders, ethnicities, residences, education levels, monthly household incomes, and regions, as well as according to families have lived in the same area for generations (refer to Figure 2). Using the significant factors identified in the univariate analysis as independent variables and the healthy diet KAP score as the dependent variable, multiple linear regression was conducted. The healthy diet KAP score of middle-aged and older adults was found to be influenced by age, gender, residence, education level, monthly household income, and region (*p* < 0.05) (refer to Table 3). In this study, compared with the ≥60 age group, the 45–59 age group exhibited higher KAP scores, probably due to lower Internet usage among individuals aged 60 and above [44]. Previous research has also indicated that the Internet serves as one of the sources for acquiring healthy diet information [45]. A study by Ma et al. [46] found that higher education level was associated with higher scores in nutritional knowledge, and research by Wang et al. [47] found that doctoral students had significantly higher scores in healthy diet KAP than undergraduate and graduate students. This study also found a positive correlation between education level and healthy diet KAP score. A study conducted in China revealed that individuals with higher levels of education scored higher in terms of salt-related knowledge as well as having higher adoption rates of healthy diet practices such as reducing salt intake [48]. This finding might be because middle-aged and older adults with high education levels have a strong interest in diet knowledge and healthy eating and may have a good understanding of nutritional requirements, food combinations, and the principles of a healthy diet. Monthly household income was identified as a factor influencing the healthy diet KAP score of middle-aged and older adults in southwestern China. This finding may have resulted from the fact that having a relatively high monthly household income means being able to buy a good variety of food and being able to enjoy a balanced and varied diet. Conversely, having a low monthly household income may lead to financial constraints and choosing affordable foods that are not nutritionally balanced. Furthermore, Yunnan, Sichuan, and Chongqing had higher scores in healthy diet KAP compared with Guizhou. This finding could be attributed to differences in dietary culture and varying levels of economic development [49].

## 5. Conclusions

The present study has some limitations. First, this study used convenience sampling to select research participants, resulting in insufficient sample representativeness. Second, this study is a cross-sectional study; hence, no causal inferences regarding the results can be made. Third, the evaluation of healthy diet-related knowledge, attitudes, and practices among middle-aged and older adults was based on self-reported data, which may be influenced by recall bias.

Overall, the results of the healthy diet KAP among middle-aged and older adults in Southwest China were somewhat unsatisfactory. Specifically, the passing rate for healthy diet knowledge was relatively low, and the passing rate for certain specific healthy diet practices was not ideal. However, there was a positive trend in the attitude towards healthy diet. Additionally, age, gender, residence, education level, monthly household income, and region were associated with healthy diet KAP score among middle-aged and older adults. The following recommendations are proposed to promote healthy diet behaviors among middle-aged and older adults: (1) Develop and implement comprehensive public health campaigns that focus on enhancing knowledge and awareness of a healthy diet among this population. Utilize various channels such as television, radio, and social media to disseminate accurate and easily understandable information about the importance of a balanced and nutritious diet. (2) Establish collaborations between government agencies, healthcare providers, and community organizations to provide specialized nutrition and health guidance services for middle-aged and older adults. These services should include personalized dietary plans to meet individual nutritional needs and provide guidance on healthy cooking methods. (3) Encourage middle-aged and older adults to actively seek out and participate in educational programs and resources related to healthy eating. Additionally, organize family dietary guidance activities to promote shared responsibility for healthy eating within households.

## Figures and Tables

**Figure 1 nutrients-16-00869-f001:**
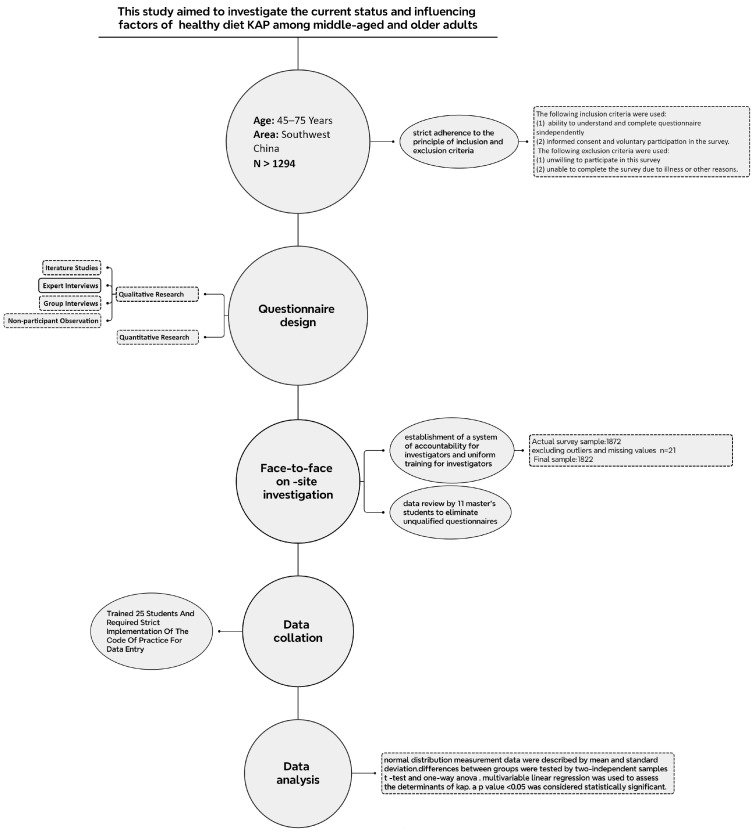
Flowchart of the cross-sectional study of KAP for healthy diets in middle-aged and older adults in Southwest China.

**Figure 2 nutrients-16-00869-f002:**
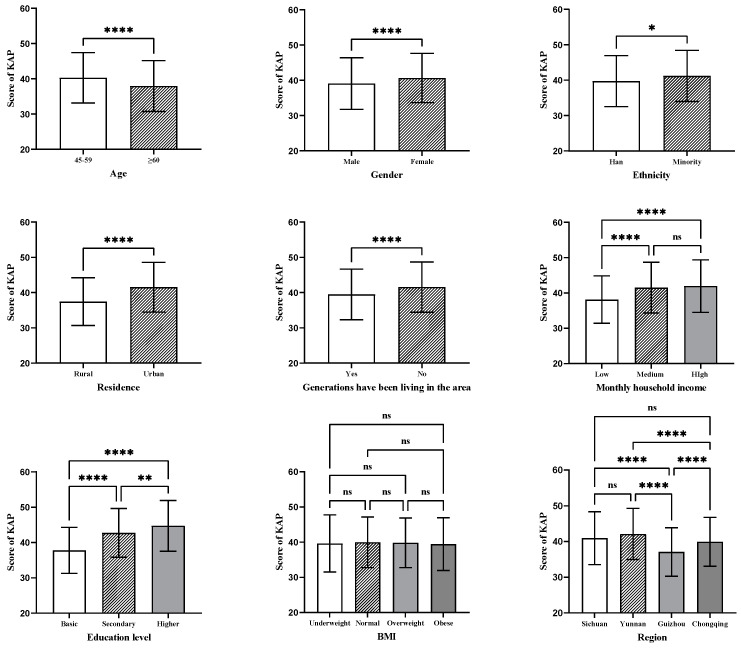
Healthy diet KAP score for different demographic characteristics. Notes: The Bonferroni test was used for multiple comparisons in the ANOVA; ns, not significant; * *p* < 0.05, ** *p* < 0.01, **** *p* < 0.0001.

**Table 1 nutrients-16-00869-t001:** General profile of the participants (*n* = 1822).

Variables		*n* (%)
Age	45–59	1473 (81.2)
	≥60	343 (18.8)
Gender	Male	914 (50.2)
	Female	908 (49.8)
Ethnicity	Han	1670 (91.7)
	Minority	152 (8.3)
Family residence	Rural	750 (41.2)
	Urban	1072 (58.8)
Generations have lived in the area	Yes	1508 (81.8)
	No	314 (17.2)
Education	Basic	1122 (61.6)
	Secondary	550 (30.2)
	Higher	150 (8.2)
Monthly household income	Low	950 (52.2)
	Medium	454 (24.9)
	High	418 (22.9)
BMI	Underweight	70 (3.8)
	Normal	1024 (56.2)
	Overweight	604 (33.2)
	Obese	124 (6.8)
Region	Guizhou	441 (24.2)
	Chongqing	677 (37.2)
	Sichuan	363 (19.9)
	Yunnan	341 (18.7)

**Table 2 nutrients-16-00869-t002:** Average score of healthy diet-related knowledge, attitudes, and practices.

Variables		Score of Knowledge	*p*-Value	Score of Attitude	*p*-Value	Score of Practice	*p*-Value
Age ^a^	45–59	4.94 ± 3.01	<0.001 *	21.54 ± 4.13	<0.001 *	13.81 ± 2.86	0.162
	≥60	4.30 ± 2.82		20.08 ± 4.19		13.57 ± 2.75	
Gender ^a^	Male	4.63 ± 2.89	0.006 *	21.11 ± 4.34	0.104	13.33 ± 2.92	<0.001 *
	Female	5.01 ± 3.06		21.42 ± 4.00		14.19 ± 2.69	
Ethnicity ^a^	Han	4.80 ± 2.97	0.435	21.19 ± 4.19	0.014 *	13.73 ± 2.86	0.094
	Minority	5.00 ± 3.19		22.06 ± 3.94		14.13 ± 2.66	
Residence ^a^	Rural	3.96 ± 2.60	<0.001 *	20.32 ± 4.38	<0.001 *	13.17 ± 2.74	<0.001 *
	Urban	5.42 ± 3.08		21.92 ± 3.89		14.18 ± 2.84	
Generations have lived in the area ^a^	Yes	4.67 ± 2.94	<0.001 *	21.16 ± 4.21	0.021 *	13.66 ± 2.83	0.001 *
	No	5.53 ± 3.11		21.76 ± 3.98		14.25 ± 2.84	
Education level ^b^	Basic	3.97 ± 2.56	<0.001 *	20.39 ± 4.15	<0.001 *	13.41 ± 2.75	<0.001 *
	Secondary	5.95 ± 3.06		22.48 ± 3.88		14.31 ± 2.86	
	Higher	7.00 ± 3.14		23.34 ± 3.53		14.39 ± 3.07	
Monthly household income ^b^	Low	4.12 ± 2.64	<0.001 *	20.45 ± 4.14	<0.001 *	13.56 ± 2.79	0.006 *
	Medium	5.44 ± 3.04		22.06 ± 4.01		13.99 ± 2.79	
	High	5.73 ± 3.26		22.24 ± 4.08		13.98 ± 2.99	
BMI ^b^	Underweight	5.31 ± 3.51	0.055	20.81 ± 4.61	0.474	13.49 ± 2.73	0.377
	Normal	4.89 ± 2.99		21.34 ± 4.08		13.70 ± 2.85	
	Overweight	4.73 ± 2.86		21.12 ± 4.26		13.92 ± 2.85	
	Obese	4.23 ± 3.09		21.56 ± 4.29		13.65 ± 2.79	
Region ^b^	Guizhou	3.57 ± 2.57	<0.001 *	20.31 ± 4.34	<0.001 *	13.19 ± 2.69	<0.001 *
	Chongqing	5.01 ± 2.90		21.18 ± 4.09		13.72 ± 2.85	
	Sichuan	5.32 ± 3.13		21.49 ± 4.22		14.13 ± 2.77	
	Yunnan	5.52 ± 3.02		22.43 ± 3.78		14.18 ± 2.98	

Notes: (1) ^a^ *t*-tests were conducted to compare the means of two groups; (2) ^b^ ANOVA was performed to assess the differences among means of multiple groups; (3) * statistical significance (*p* < 0.05).

**Table 3 nutrients-16-00869-t003:** Multiple linear regressions to identify factors affecting healthy diet KAP.

Variables	*β*	*SE*	Beta	*t*	*p*-Value
Age					
45–59 (Ref)					
≥60	−1.333	0.387	−0.072	−3.446	0.001 *
Gender					
Male (Ref)					
Female	1.808	0.302	0.125	5.983	<0.001 *
Ethnicity					
Han (Ref)					
Minority	1.083	0.557	0.042	1.943	0.052
Residence					
Rural (Ref)					
Urban	1.945	0.340	0.133	5.719	<0.001 *
Generations have lived in the area					
Yes (Ref)					
No	0.694	0.412	0.036	1.684	0.092
Education level					
Basic (Ref)					
Secondary	3.493	0.367	0.223	9.522	<0.001 *
Higher	4.830	0.613	0.184	7.875	<0.001 *
Monthly household income					
Low (Ref)					
Medium	1.330	0.386	0.080	3.443	0.001 *
High	0.923	0.410	0.054	2.254	0.024 *
Region					
Guizhou (Ref)					
Chongqing	2.259	0.399	0.151	5.661	<0.001 *
Sichuan	3.092	0.464	0.171	6.660	<0.001 *
Yunnan	3.153	0.475	0.171	6.640	<0.001 *

Notes: Adjusted for each other (when exploring the relationship between one demographic factor and the healthy diet KAP, the others were adjusted); * statistical significance (*p* < 0.05).

## Data Availability

The datasets generated and/or analyzed during the current study are not publicly available due to funding requirements but are available from the corresponding author on reasonable request.

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
