# Peer review of "Healthy Diet-Related Knowledge, Attitude, and Practice (KAP) and Related Socio-Demographic Characteristics among Middle-Aged and Older Adults: A Cross-Sectional Survey in Southwest China"

_nutrients, 2024, doi:10.3390/nu16060869_

Round 1

Reviewer 1 Report

Comments and Suggestions for Authors

Dear Authors,

There are several improvements needed to enhance the overall quality of the manuscript. My comments and suggestions are included below:

The introduction fails to sufficiently explain the research context. How the KAP model can be relevant for this study should be better explained.

L 77 “Therefore, this study primarily aimed to investigate the KAP of healthy diet among middle-aged and older adults in southwestern China “ -The objective could be more clearly defined. To enhance clarity, it would be beneficial to specify the specific aspects of the KAP model that the study seeks to explore and the intended outcomes or implications of the research.

The structure of the questionnaire should be described more in detail particularly by delineating how each section corresponds to the Knowledge-Attitude-Practice (KAP) model.

L 132 “Additionally, the pre-survey results showed that Cronbach’s alpha coefficient was 0.825, indicating that the questionnaire had acceptable reliability” – it lacks clarity. Provide more details on pre-survey role in the development and validation of research measurement instrument.

The presentation of results lacks the necessary level of detail to thoroughly understand the methodology used to generate the results and assess their validity.

L 288 “The level of healthy diet KAP among middle-aged and older adults in southwestern China was unsatisfactory. At the same time, they had inadequate knowledge and inappropriate behavior toward healthy diet” - the conclusions go beyond the results presented.

Overall, the article needs expansion to clarify methodological aspects related to both the research instrument and its connection to the KAP model.

Reviewer 2 Report

Comments and Suggestions for Authors

The authors investigated the current status and influencing factors of healthy diet knowledge, attitude, and practice (KAP) among middle-to-older adults between ages 45–75 years in Southwest China. The findings appear very promising, and seem well discussed, too. The authors are encouraged to consider the following:
a) The introduction needs additional information. The end of paragraph one should include short-to-long term implications of lack of nutrition health-related knowledge to a given population. The beginning of paragraph 2 should start with the overall health situation of middle-to-elderly aged groups in Asia (make sure to cite China and other Asian nations), then, how the greater population of China makes the situation highly important, especially from the regional context. Then, talk about China's health-nutrition intake implications, and the importance of understanding food types vis-a-vis influential factors.In the last paragraph, tell us why this study objective is very important, and the hypothesis/research question that was raised to justify it

b) Materials and methods needs additional work, and further amendments/additions. Firstly, please create a new subsection to start the materials and methods, titled "Schematic overview of the cross-sectional survey", which must comprise not less than 4 sentences, and a detailed flow diagram that shows the major steps followed in the entire study. Suggestion of flow can be: Identification of research survey objective>  Elucidation of survey contents/elements>Development of questionnaire survey /design strategy> Sample size and participation> Questionnaire administration process> Data collation and analysis
Two sentences should be said about this flow diagram, and make sure it corroborates with the study objective, and the raised hypothesis. The reviewer will examine this section critically
Subsequent sections in the methodology should follow the flow chat key steps ok
c)Results and discussion are ok. Please authors, kindly make effort to eliminate flowery words in the discussion. Be concise and direct in the discussion. Also, as many places where the data from all tables and figures are mentioned in the discussion, please make sure to put "(Refer to Table x)" or "(Refer to Figure x)". So, all the table/figure in the results section must be captured in the discussion.

Please end of the discussion, remove the limitations, and put them in the conclusions. Let it start the conclusions ok. Change the title 'Conclusions' to "Concluding remarks" (It tends to fit better for survey questionnaire based studies of this nature)

Make every effort to incorporate to the best detail all the above. You will be very happy and proud of the revised manuscript when you accomplish all these. Look forward to your revised manuscript.

Reviewer 3 Report

Comments and Suggestions for Authors

Dear Authors,

Ever since China embarked on the path of economic transition and rapid economic growth, the dietary and lifestyle patterns of its people have been changing. However, the changes are going in an undesirable direction, since the percentage of overweight and obese people in the country's population is increasing dynamically (in the sample studied, they accounted for 40%), since the government and local authorities have to face the new problem of the double burden, and since data from international institutions such as FAO, WHO, etc. show a dynamic increase in the consumption of meat, sugar and an increase in the proportion of energy from fat and free sugars in the average diet. In this context, the topic of the reviewed manuscript is important so that, based on the results of the study, various types of educational activities and nutritional interventions can be implemented, as it may not be too late yet.

Firstly, I suggest to avoid in the introduction the vague and long-standing scientific facts about the relationship between diet and risk of NCDs, which apply to all age groups of the population ((as for example in the first paragraph of L.38-50) .  And if the authors want to write about an ageing population, data for China should be presented in addition to the data from source [2]. Furthermore, I suggest supplementing the introduction with information/data on the health status of the Chinese population or in the SWC, how many inhabitants the region has, what a typical conventional diet looks like - food raw materials, nutritional value, seasonal differences in eating habits, etc. and how this diet has changed over the last 20 years of economic development and China's WTO membership. Please also explain what it is about the geographical environment of the SWC that influences the dietary preferences of the people in the region (L. 51).

Secondly, the Data Collection section needs to be supplemented with the following information: the method of sampling (we know from the Limitations section that it was convenient, but this is not enough); what technique was used to conduct the survey (CAWI? PAPI? etc.); we also know that the sample was not representative, but perhaps proportionality (quota selection) was adopted in relation to some socio-demographic characteristic of the respondents.

Thirdly, regarding the poor and unsatisfactory results for the KAP model, the Discussion section should include specific proposals for educational activities and nutrition interventions (e.g. in educational institutions and schools) based on successful health initiatives and strategies in other countries. The issues outlined on page 8 have already taken place in Western countries, so China's nutrition policy actions should benefit from their experience. I also suggest that in this chapter, the research results of other authors cited in the manuscript should be presented in a table, then their perception will be much higher. There are 10 publications in the list of references where the KAP model was used.

Fourthly, the questionnaire is in Chinese, so it should be translated into English and posted on the MDPI website in that version.

Kind regard

Reviewer 4 Report

Comments and Suggestions for Authors

Title, Abstract and Keywords:

The title of the paper is eel indicative of the research described in the manuscript text.

The abstract is very well organized, providing some contextualization stating the aims of the study, next presenting some major findings and ends with a brief concluding remark and recommendation.

Introduction:

The introduction helps to frame and contextualize the work based on a supporting Literature Review The justification of study is provided at the end of introduction along with the objectives.

Materials and Methods

This part is generally correct, but the statistical part is not detailed enough. TFor example significant differences with T-test mean a difference between two mean values. However when ANOVA is applied it is not possible to know where the differences stand, and therefore it is mandatory to complement ANOVA with some kind of post hoc test to evaluate which means are effectively different form each other.

Also, when performing a multiple regression analysis, it is mandatory to obey the necessary conditions, i.e., this type of analysis can only be performed if some assumptions are verified, such as: a) normality of the errors, b) constant variance of the errors, c) null mean of the errors (is a consequence if the least squares method is used), d) independency of the errors and e) absence of multicolinearity.

Additionally it is imperative to perform the test F to the significance of the regression (not only verify the significance of the coefficients of each variable independently).

These aspects are not highlighted in the M&M part and not in the results either, and therefore, it is not verified if utilization of multiple linear regression is applicable to this set of data. An experienced statistician should help complete the analyses and verify the assumptions.

Results

In table 2, next to each variable a footnote must be added specifying if the T-test or ANOVA were used and the significance level: Ex: Age1, Education Level2 – when ANOVA is used the Post-Hoc used must be mentioned.

For the cases of ANOVA, the mean values must be accompanied with appropriate superscripts (a, b, c..) to identify significant differences, and in the footnote the meaning of the superscripts must be clarified.

In the presentation of results in Table 3, for example in the text evidence must be provided of the fulfilment of the assumptions necessary to apply multiple regression analysis.

Discussion

This part is generally OK, although it could have been better explored.

Conclusions:

This part is too synthetic, and more evidence should be given to the different conclusions and not have only a very general  overview.

Comments on the Quality of English Language

Title, Abstract and Keywords:

The title of the paper is eel indicative of the research described in the manuscript text.

The abstract is very well organized, providing some contextualization stating the aims of the study, next presenting some major findings and ends with a brief concluding remark and recommendation.

Introduction:

The introduction helps to frame and contextualize the work based on a supporting Literature Review The justification of study is provided at the end of introduction along with the objectives.

Materials and Methods

This part is generally correct, but the statistical part is not detailed enough. TFor example significant differences with T-test mean a difference between two mean values. However when ANOVA is applied it is not possible to know where the differences stand, and therefore it is mandatory to complement ANOVA with some kind of post hoc test to evaluate which means are effectively different form each other.

Also, when performing a multiple regression analysis, it is mandatory to obey the necessary conditions, i.e., this type of analysis can only be performed if some assumptions are verified, such as: a) normality of the errors, b) constant variance of the errors, c) null mean of the errors (is a consequence if the least squares method is used), d) independency of the errors and e) absence of multicolinearity.

Additionally it is imperative to perform the test F to the significance of the regression (not only verify the significance of the coefficients of each variable independently).

These aspects are not highlighted in the M&M part and not in the results either, and therefore, it is not verified if utilization of multiple linear regression is applicable to this set of data. An experienced statistician should help complete the analyses and verify the assumptions.

Results

In table 2, next to each variable a footnote must be added specifying if the T-test or ANOVA were used and the significance level: Ex: Age1, Education Level2 – when ANOVA is used the Post-Hoc used must be mentioned.

For the cases of ANOVA, the mean values must be accompanied with appropriate superscripts (a, b, c..) to identify significant differences, and in the footnote the meaning of the superscripts must be clarified.

In the presentation of results in Table 3, for example in the text evidence must be provided of the fulfilment of the assumptions necessary to apply multiple regression analysis.

Discussion

This part is generally OK, although it could have been better explored.

Conclusions:

This part is too synthetic, and more evidence should be given to the different conclusions and not have only a very general  overview.

Round 2

Reviewer 1 Report

Comments and Suggestions for Authors

Dear Authors,

The manuscript has been improved, but as I mentioned previously, there are some issues pertaining to the methodology. For greater clarity, you should elaborate on how the KAP scores were interpreted. Consequently, it is necessary to include research hypotheses. To enhance understanding of the study's Chinese context, the discussion should more closely relate your findings to existing research focused on China.

Reviewer 3 Report

Comments and Suggestions for Authors

Dear Authors,
thank you for your comprehensive responses to my review. I appreciate the additions included in the revised version of the manuscript as suggested by me. I like the flow chart of the study, including the information regarding the sampling and conduct of the study that I requested. The only information still missing is - what method was used to conduct the study? - people aged 75 are unlikely to take part in online surveys. Was it therefore a face-to-face survey? Please add this information.

Kind regards

Reviewer 4 Report

Comments and Suggestions for Authors

I recognize that the autors did a good effort to improve their work and I believe now it can be acepted for publication.

Comments on the Quality of English Language

Genereally acceptabel.
